# Circularly polarised luminescence laser scanning confocal microscopy to study live cell chiral molecular interactions

Patrycja Stachelek [1,2], Lewis MacKenzie [1,2], David Parker [1] & Robert Pal[1✉]

The molecular machinery of life is founded on chiral building blocks, but no experimental technique is currently available to distinguish or monitor chiral systems in live cell bio-imaging studies. Luminescent chiral molecules encode a unique optical fingerprint within emitted circularly polarized light (CPL) carrying information about the molecular environment, conformation, and binding state. Here, we present a CPL Laser Scanning Confocal Microscope (CPL-LSCM) capable of simultaneous chiroptical contrast based live-cell imaging of endogenous and engineered CPL-active cellular probes. Further, we demonstrate that CPL-active probes can be activated using two-photon excitation, with complete CPL spectrum recovery. The combination of these two milestone results empowers the multi-disciplinary imaging community, allowing the study of chiral interactions on a sub-cellular level in a new (chiral) light.

[1] Department of Chemistry, Durham University, South Road, Durham DH1 3LE, UK. [2] These authors contributed equally: Patrycja Stachelek, Lewis MacKenzie ✉email: robert.pal@durham.ac.uk

Chirality is intrinsic to the fundamentals of life, with chiral molecules playing pivotal roles in natural cell homoeostasis, chemical signalling, enzymatic reactions, gene expression and reproduction[1]. Selective chiral recognition of vital bio-molecules and bioprocesses by specifically designed probes in biological environments, like live cells, is a topic of great interest from both biochemical and analytical application points of view. Tracking chiral molecules within live cells can be of great importance to understanding interactions between cell organelles and drugs or chiral probes. It has been shown that chirality can have a profound effect on drug activity and on localisation in cells amongst others. Stereoselectivity, particularly enantioselectivity is of crucial importance when designing either drugs and/ or molecular probes to interact with proteins[2,3]. Despite life being founded on chiral building blocks, currently, there are no direct techniques capable of distinguishing and tracking the behaviour of enantiomers in chiral molecular systems—e.g. luminescent probes or proteins—and their individual interactions within living cells.

Circular polarisation luminescence (CPL) spectroscopy and circular dichroism (CD) spectroscopy are optical spectroscopic tools for studying chiral molecules and their interactions, but to date, these have not met the complex biocompatible requirements needed to study and track chiral interactions within cells. CD spectroscopy quantifies how chiral molecules absorb left and right circularly polarised light differentially. CD microscopes have been developed and applied to applications such as the microscopy of inorganic chiral substrates and structures, but CD microscopes face several limitations, including artefacts associated with the linearly polarised light components and optical distortion[4–7]. Fundamentally, CD techniques are limited by the optical absorbance of the sample, rendering CD too insensitive and of little use for live-cell imaging.

On the other hand, CPL spectroscopy enables the study of chiral interactions of luminescent molecular systems. Efficient CPL probe complexes, foremost lanthanide coordination complexes, can be used for cellular imaging because of their demonstrated biocompatibility and unique photophysical and CPL properties[8]. The chirality of such complexes and their binding motifs enables them to be exploited as sub-cellular chiral probes[9]. However, CPL spectroscopy and measurement technology has lagged behind the development of CPL-active probes, hindering the translation to sub-cellular CPL microscopy. Since the inception of CPL spectrometers roughly 50 years ago photoelastic modulators (PEMs) have been incorporated for circular polarisation analysis, in combination with scanning monochromators for wavelength specificity. Such setups have resulted in bulky, expensive instruments with slow spectral data acquisition rates (~1 h for a typical CPL spectrum)[10,11]. Consequently, the few CPL microscopes which have been developed incorporate PEMs, and they, therefore, suffer from inherently slow-acquisition rates which are incompatible with imaging CPL within living cells due to: (a) high-energy excitation (e.g. UV/ blue) induced phototoxicity and (b) correspondingly poor spatial resolution owing to increased acquisition times. Consequently, circularly polarised emission microscopy studies have been limited to investigations of inert, bulk samples at the millimetre scale, e.g. large inorganic crystals[12], or imaging maps of chiral amino acids distributed within agarose gel[13]. This situation is in stark contrast to the many advances in optical microscopy over recent decades, which have enabled diffraction-limited and super-resolved cellular imaging[14].

To advance CPL microscopy, we have previously built a proof-of-concept chiroptical contrast time-resolved epifluorescence microscope and demonstrated enantioselective differential chiral contrast (EDCC) imaging, examining the Λ- and Δ-enantiomers of a bright CPL-active europium complex (Eu:L1, Eu[1,4,7- tris({4-[2-(4-methoxy-2- methylphenyl)ethynyl]-6-[carboxy(phenyl)phosphoryl] pyridin-2-yl}methyl)-1,4,7- triazacyclononane]) (structure depicted on Fig. 3) absorbed onto an optical brightener-free paper test substrate[9]. This study showed that the enantiomers could be spatially discriminated by their CPL emission profile. This epifluorescence CPL microscope was considerably faster than PEM-CPL microscopes (2 min vs. 40 min total image acquisition), but still required sequential manual selection of CPL detection channels, which is impractical for sub-cellular imaging where high spatio-temporal resolution is necessary to track chiral interactions without the introduction of image artefacts. A microscope capable of simultaneously measuring left and right-handed CPL (ideally with confocal sectioning and truly diffraction-limited resolution) represents a step-change in technological capability, opening up new opportunities to study chiral molecular interactions.

We recently introduced next-generation CPL spectrometer technology which circumvents the aforementioned hindrances of 20th century CPL spectroscopy technology, using a patented optical layout and rapid charge-coupled device (CCD) spectrometers[15].

Here, we demonstrate the extension of this rapid CPL measurement technology to CPL laser scanning confocal microscopy (CPL-LSCM). This approach allows the simultaneous acquisition of left and right-handed CPL images for enantioselective differential contrast microscopy, with a diffraction-limited spatial resolution (i.e. 126 nm lateral, 396 nm axial resolution using ×63 1.4 NA (numerical aperture) objective) across a typical field of view (FOV) of 100 × 100 µm in 9 s, when employing 355 nm single-photon activation with an Nd:YAG (3rd harmonic) laser. The CPL-LSCM could be a radical new tool in bio-imaging enabling the study of fundamental chiral interactions within live cells by harnessing the unique and uncharted power of CPL. In addition to cellular applications, CPL-LSCM could be used in studies of thin and thick film embedded CPL emitters, for applications such as physically unclonable multi-layered CPL-active security inks[16,17], and next-generation organic light-emitting diodes[18].

## Results

Throughout this work when discussing europium CPL, we selected the magnetic dipole (MD) allowed $\Delta J = 1$ transition ($\lambda \sim 595$ nm) because it exhibits both high overall emission intensity and a high $g_{em}$ value (i.e. an overall high CPL brightness) for the employed Eu:complexes. In this MD allowed transition manifold around 595 nm, europium complexes with transitions of the same sign of CPL emission have been identified. Therefore, using optical filters, the whole $\Delta J = 1$ manifold can be selected maximising light collection efficiency; it represents the best emission window to examine for Eu complexes in CPL-LSCM.

**Multi-photon (MP) CPL spectroscopy.** CPL spectroscopy is a well-established discipline, with CPL emission maximised for emission transitions which are MD allowed and electric dipole (ED) forbidden. CPL emission is most commonly quantified in terms of the emission dissymmetry factor, $g_{em}$, calculated by

$$g_{em} = \frac{2(I_{L-CPL} - I_{R-CPL})}{(I_{L-CPL} + I_{R-CPL})}. \quad (1)$$

Where $I_{(L-CPL)}$ and $I_{(R-CPL)}$ are the intensities of (left-handed) L-CPL and (right-handed) R-CPL emission; $g_{em} = 2$ indicates 100% L-CPL emission, $g_{em} = -2$ indicates 100% R-CPL emission, and $g_{em} = 0$ indicates net-zero circular polarisation. The strongest CPL signals to date have been generated by lanthanide coordination complexes, specifically europium complexes, exhibiting $g_{em} = |1.38|$ for single complexes and $g_{em} = |1.45|$ for

chiral supramolecular polymers[19,20]. Recently, the more comprehensive metric of circularly polarised brightness (CPB) has been introduced[16,21]. CPB serves as a metric for the total number of circularly polarised photons emitted and combines the molar absorption coefficient ($\xi_{abs}$), quantum yield ($\phi_{em}$), and $g_{em}$. CPB is calculated as

$$CPB = \xi_{abs}\phi_{em}(g_{em}/2) \qquad (2)$$

It has units of $cm^{-1}M^{-1}$. A wide variety of CPL emitting molecular systems can produce useable CPB values (loosely defined here as ~>50 $cm^{-1}M^{-1}$), including BODIPYs, helicenes, excimers, cyclophanes, and d-metal complexes[21]. However, for the purposes of CPL-LSCM, chiral lanthanide complexes are pre-eminent owing to their excellent photophysical properties: large Stokes' shifts, narrow, line-like emission spectra, tunable emission and excitation properties, intricate CPL emission, with the potential for exceptional $g_{em}$ and CPB values (e.g. CPB values of >1000 and >3000 $cm^{-1}M^{-1}$ with europium complexes)[5]. However, CPL spectroscopy to date has been concerned with single-photon excitation, which is sub-optimal for cellular imaging studies due to the associated potential photo-damage and shallow tissue penetration of near-UV excitation.

Multiphoton (MP) spectroscopy and microscopy techniques reduce cellular phototoxicity by harnessing a tightly focused low-energy near infra-red (NIR) laser pulse (typically femtoseconds) to stimulate emission from fluorescent/luminescent molecules that would otherwise require absorption of a single high-energy UV photon[22]. The tightly defined spatial constraints of MP microscopy offers both improved axial (z-axis) resolution and sub-cellular sectioning capability for imaging and photodynamic therapy applications[23]. However, the field of MP-CPL has been largely unexplored. We are aware of just one previous report of an MP up-conversion-CPL spectrum in the literature of chiral perovskite nanocrystals[24].

Two-photon excitation (2PE) CPL spectroscopic studies of a europium (III) complex were undertaken by coupling a tunable femtosecond pulsed laser (680–1300 nm, Coherent Discovery TPC, 100 fs, 80 MHz) to two pre-existing CPL spectrometers. (1) a conventional PEM[25] CPL spectrometer, and (2) a next-generation solid-state CCD (SS-CCD) CPL spectrometer[15]. Either approach yielded the same CPL spectrum, measuring a model CPL complex, modified Eu:BPEPC (Eu[[(4-methoxyphenyl)alkynyl-bpepc]₃}Cl₃ Fig. 1)[26], but as per previous publications, the SS-CCD-CPL spectrometer acquired the CPL spectrum 800 times faster than the PEM-CPL spectrometer (under 10 s for a 1000 accumulation spectrum). For the purposes of this work, we will report 2PE-CPL spectra from the PEM CPL spectrometer because this form of CPL spectrometry represents the current benchmark instrumentation of the CPL spectroscopy community.

For efficient two-photon excitation, a material must have a high two-photon absorption (TPA, 2PA) cross-section ($\sigma_2$) (GM) with a favourable emission quantum yield. Therefore, we extend CPB to two-photon-activation CPB (2PE-CPB or $CPB_{2PE}$) as

$$2PE - CPB = \sigma^2 \times \phi_{em} \times \frac{g_{em}}{2} \qquad (3)$$

Certain lanthanide complexes (especially those of Eu(III) and Tb(III)) have adequate two-photon cross-sections[27], leading to good 2PE-CPB values, paving the way towards two-photon activated live-cell CPL-LSCM. We determined the cross-sections ($\sigma^2$), of the standard materials (modified Eu:BPEPC and Eu:L1) discussed herein[26,28], according to established procedures (see SI)[27,29]. It is important to note that due to the nonlinear effect of 2PE (non-degenerate two-photon absorption) the 2PE wavelength is often lower and extremely rarely double that of the 1PE absorption maximum. Due to the

quadratic relationship between the intensity of the 2PE excitation and the triggering of a fluorescent event the shape of 2PE excitation is inherently always far narrower, sharper too than that of the 1PE excitation[23,29].

We verified that the MP excitation process was a two-photon event by recording an excitation power dependence; confirmed by the resulting line having a slope of ~2 on a logarithmic scale (see Fig. 1A). We measured the MP cross-section of the recently developed CPL-standard Eu(III)-complex, (modified Eu:BPEPC) via these methods (Fig. 1). Its 2PE cross-section was determined to be $\sigma^2 = 142 \pm 3$ and $147 \pm 3$ GM (1 GM = $10^{-50}$ $cm^4$ s photon$^{-1}$) for the Λ- and Δ-enantiomers in methanol (MeOH), respectively. This result is particularly notable because MP cross-sections have not previously been determined for enantiomers of CPL active materials. Eu:L1, is a complex which has shown enantioselective localisation in live-cell experiments[27,30]. The MP cross-section of Eu:L1 in MeOH was determined to be $\sigma^2 = 51 \pm 3$ GM for both the Λ- and Δ-enantiomers (see SI Fig. 1).

**Enantioselective CPL-LSCM microscopy imaging.** Details of the CPL-LSCM set-up are shown in Fig. 2. In brief, light from the sample plane of the LSCM exits the microscope via an output port and into an external CPL analysis module, which is adapted from the rapid CPL spectrometer that we recently reported[15]. First, the waveband of interest is selected by a switchable band-pass filter. Then, an achromatic wave plate converts left and right circularly polarised light into orthogonal linearly polarised states. The light is then split into two analysis pathways by a 50:50 beam-splitter cube. The two linear polarised light states generated (horizontal and vertical polarisation) corresponding to left or right CPL are selected by a carefully aligned linear polarizer, housed in a high-precision computer-controlled rotation mount[15]. The emission intensity of each pixel is quantified in a conventional LSCM scanned manner by a dedicated high sensitivity avalanche photodiode pair. Whilst each detection arm can operate independently, both components are matched in alignment and specification to enable rapid and simultaneous acquisition of left and right CPL images. Full technical details of the CPL-LSCM system and the applied image processing methodology are provided in the Supporting information.

EDCC imaging was realised by subtraction of the simultaneously recorded left-handed CPL image from the right-handed CPL (and vice versa) using ImageJ software (v1.49)[31].

To demonstrate the CPL-LSCM on well-controlled targets, we first re-created the paper test-target [Test Target 1] which had been used for the proof-of-concept CPL microscopy studies reported in 2016[9]. This was achieved by simply depositing 15 μL of enantiopure solutions of Eu:L1 ($10^{-4}$ M) in MeOH onto the commercially available optical brightener-free paper substrate from Canson®, allowing the sample to dry at room temperature. This procedure replicated the results obtained with the epifluorescence microscope and further demonstrated the excellent performance of the CPL-LSCM. The experiment revealed differences arising from the texture of the brightener-free paper. Under the applied ×40 (0.7 NA, air) magnification, it was found that the paper has a 'rough' and 'smooth' side, which was not observed in prior studies with the epifluorescence CPL microscope (see SI Figs. 4–6 and 8)[9]. To remove the variables introduced by the paper substrate, another test-target was developed [Test Target 2] where enantiopure solutions of Eu:L1 ($5 \times 10^{-5}$ M) were deposited into a uniform polymer matrix (Polyvinylpyrrolidone (PVP-40), Sigma-Aldrich)[9,28] by spin-coating on glass substrates (170 μm thick standard microscope coverslips). Distribution of the complexes within a polymer matrix is of particular interest for ultra-secure anti-counterfeiting

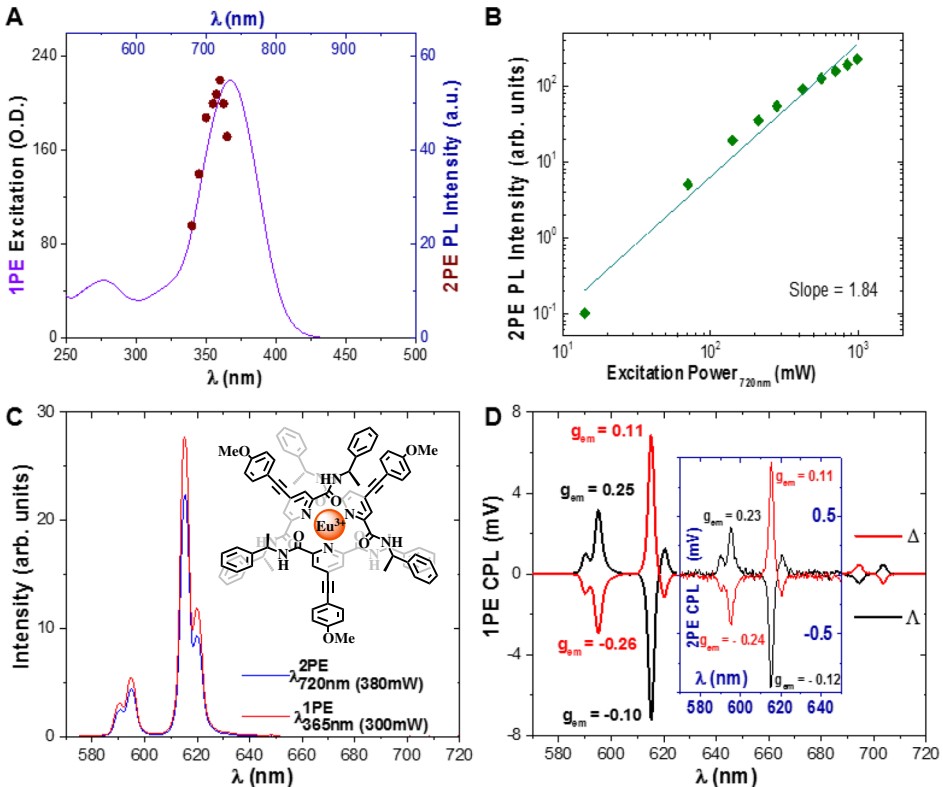

**Fig. 1 Key photophysical parameters and spectra of Λ- and Δ-modified Eu:BPEPC in MeOH. A** One photon excitation (solid purple line) and two-photon excitation (maroon dots) spectra ($\lambda_{em}$ = 615 nm) of modified Eu:BPEPC. **B** Excitation power dependency (green diamonds) of the 2PE induced photo luminescence (PL) intensity, slope 1.84 ± 0.1, $\sigma^2$ = 144.5 ± 3 GM ($10^{-50}$ cm$^4$s/photon). **C** One ($\lambda_{ex}$ = 365 nm, solid red line) and two-photon ($\lambda_{ex}$ = 720 nm, solid blue line) induced emission spectrum of the depicted (insert) Λ- and Δ Eu:BPEPC. **D** One photon and (insert) two-photon CPL spectra of Λ- and Δ-modified Eu:BPEPC (solid black and red line respectively). Spectra recorded in MeOH.

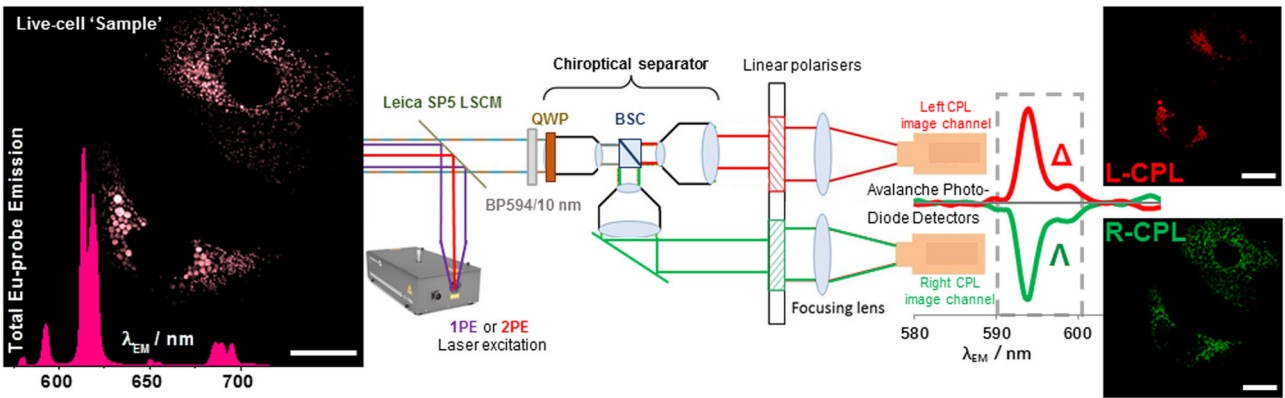

**Fig. 2 Simplified depiction of the CPL-LSCM developed for enantioselective differential chiral contrast (EDCC) imaging.** The external CPL-LCSM module is attached to a commercial LSCM via a dedicated external (X1) port to facilitate simultaneous parallel diffraction-limited enantioselective imaging of CPL-active probes within live cells. Scale bars = 20 μm.

CPL-active security ink applications (e.g. for application within plastic banknotes) and solution-processed CPL-OLEDs.

To demonstrate that Eu:L1 retains its distinctive emission properties within the PVP40 polymer, we measured CPL emission, 2PE-CPL emission, and 2PE-cross-section measurements of Eu:L1 in anhydrous N-methyl-2-pyrrolidone (NMP) (see Fig. 3). NMP was selected because NMP is structurally almost identical to the monomer unit of PVP polymers (N-Vinylpyrrolidone). This is crucial because it has recently been reported that CPL signal intensity and even CPL sign may vary as a function of solvent or medium polarity[32]. We found that the CPL signal intensity ($|g_{em}|$ and CPB) and sign were

unaltered and the 2PE cross-section of Eu:L1 was even slightly enhanced in NMP ($\sigma^2$ = 73 ± 3 GM for Λ- and Δ-enantiomer) compared to MeOH ($\sigma^2$ = 51 ± 3 GM for Λ- and Δ-enantiomer), validating our choice of host polymer for spin-coated test targets.

However, the non-uniform edges of the glass substrates caused light helicity inversion at the edges of the glass substrates owing to unwanted internal reflection, making them less than ideal for side-by-side demonstration of enantioselective CPL-LSCM. This behaviour is ascribed to dramatically reduced $g_{em}$ values observed from Test Target 2 as a result of reflected light-induced CPL helicity inversion (see SI Figs. 7 and 8).

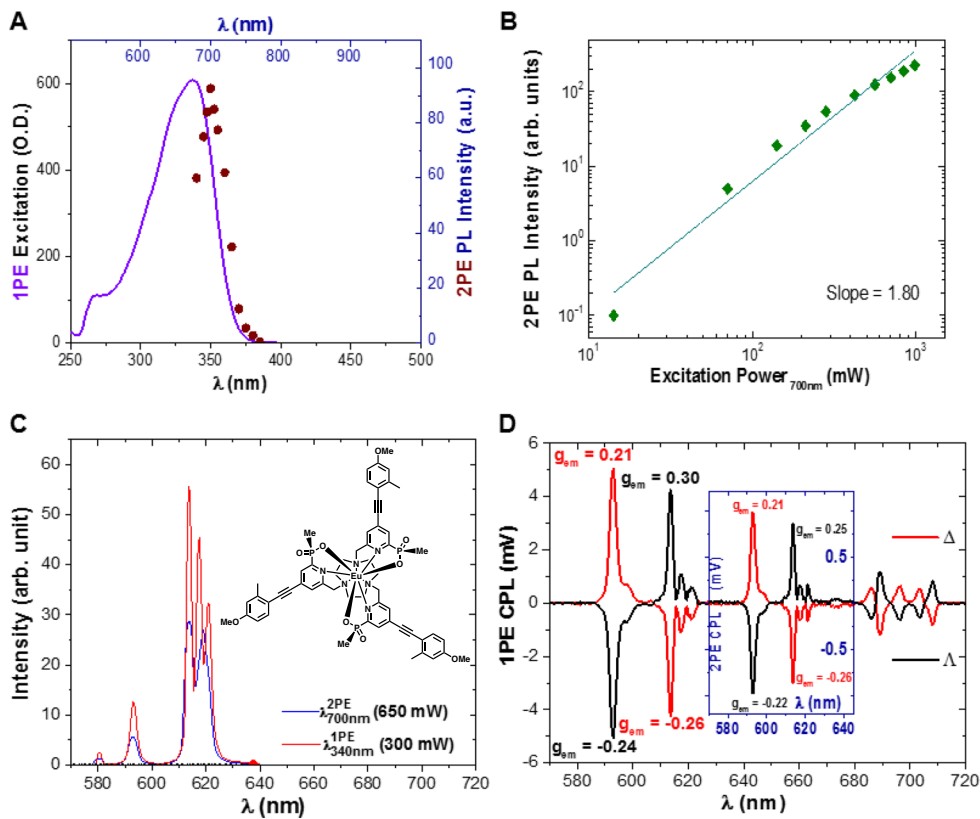

**Fig. 3 Key photophysical parameters and spectra of Λ- and Δ Eu:L1 in NMP. A** One photon excitation ($\lambda_{em} = 615$ nm) (solid purple line) and two-photon excitation (maroon dots) spectra ($\lambda_{em} = 615$ nm) of Eu:L1. **B** Excitation power dependency (green diamonds) of the 2PE induced photoluminescence (PL) intensity, slope $1.80 \pm 0.1$, $\sigma^2 = 73 \pm 3$ GM ($10^{-50}$ cm$^4$s/photon). **C** One ($\lambda_{ex} = 340$ nm, solid red line) and two-photon induced ($\lambda_{ex} = 700$ nm, solid blue line) emission spectrum of the depicted (insert) Λ- and Δ Eu:L1. **D** One photon (main figure) and (insert) two-photon CPL spectra of Λ- and Δ Eu:L1 (solid black and red line, respectively). Spectra recorded in NMP.

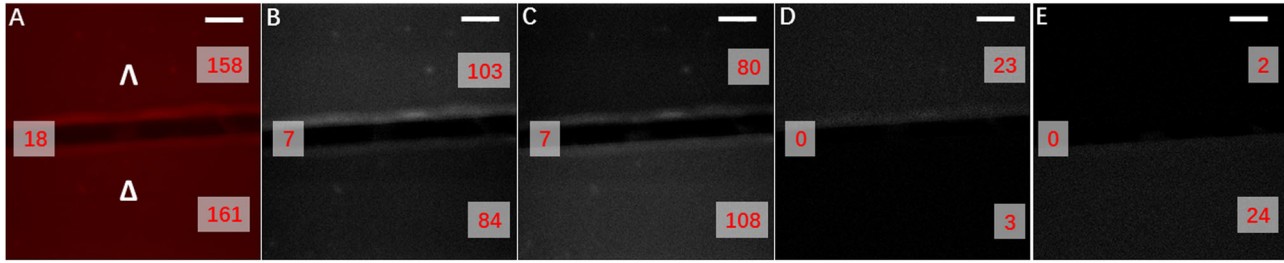

**Fig. 4 Enantioselective differential chiral contrast (EDCC) CPL-LSCM of Λ- and Δ-Eu:L1 on a glass substrate. A** Total europium emission ($\lambda_{ex} = 355$ nm, 20 mW, $\lambda_{em} = 589$ to 720 nm). **B**, **C** Left and Right Handed CPL channel respectively ($\lambda_{em} = 589$ to 599 nm). **D** Left-handed EDCC image (left CPL—right CPL). **E** Right-handed EDCC image (right CPL—left CPL). The objective used: ×40 0.7 NA air, 210 × 210 μm FOV, 100 avg, 1.5 μm axial section. Scale bars = 30 μm, numbers in red are avg. Eight-bit pixel intensity values for each image region.

A third test target was therefore developed with improved properties [Test Target 3]. This was achieved by heating a glass microscope slide to 35 °C on a hot plate, and drop-casting 100 μL of solutions of the Λ- and Δ-enantiomers ($10^{-4}$ M) 1:1 v/v Eu:L1:PVP40 solutions in MeOH side-by-side onto the slide using a multi-channel pipette. The slide was dried on the hot plate for 1 min, removed and allowed to dry at room temperature for 5 min[9].

Test Target 3 was successfully used during validation of the CPL-LSCM setup and provided superior quality diffraction-limited images (see Fig. 4 and SI Fig. 7). Simultaneous enantioselective imaging was thereby achieved in the same FOV using bright CPB CPL emitters in a validated and controlled test target of enantiopure lanthanide complexes as homogenous thin film deposits (from a 210 × 210 × 2 μm FOV). Additionally, we

measured the total CPL spectrum of the Test Target 3 using the PEM-CPL spectrometer (see SI Figs. 2 and 3), showing excellent correspondence with respect to both CPL sign and CPB.

**Enantioselective CPL-LSCM of live cells.** To demonstrate the live-cell imaging capabilities of the newly constructed CPL-LSCM, NIH 3T3 (mouse skin fibroblast, ATCC-CRL-1658) cells were dosed with 30 μM of enantiopure Λ- and Δ-Eu:L1 complex for 14 h as per Frawley et al.[28] The images in Fig. 5 demonstrate that CPL-LSCM can simultaneously collect images and differentiate between the left- and right-handed CPL emission arising from the cells. The enantiomers of Λ- and Δ-Eu:L1 appeared to localise to different organelles with well-resolved brightness differences observable within NIH 3T3 cells.

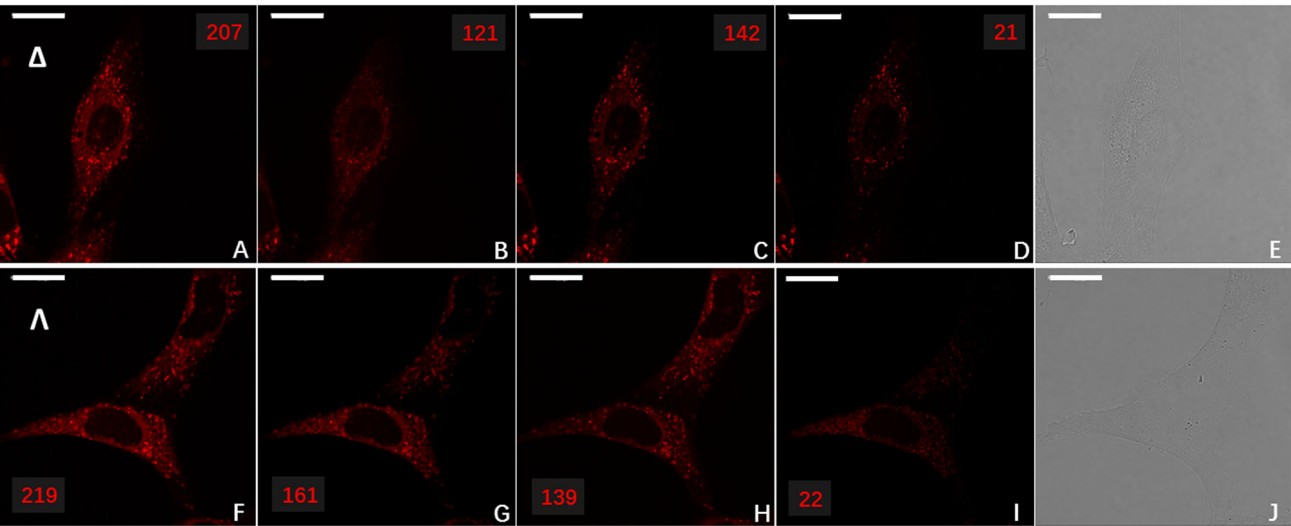

**Fig. 5 Enantioselective localisation of enantiopure europium complex, Eu:L1 to the lysosome or mitochondria in live NIH 3T3 cells.** Enantioselective differential chiral contrast (EDCC) CPL-LSCM of Eu:L1 (30 μM, 14 h loading, ×63 1.4 NA oil objective, 96 × 96 μm FOV, 100 avg., 790 nm axial section) in live mouse skin fibroblast (NIH 3T3) cells, showing enantioselective localisation to the lysosome (Δ-Eu:L1) and mitochondria (Λ-Eu:L1). Top row Δ-Eu:L1. **A** Total Europium emission ($\lambda_{ex} = 355$ nm, 20 mW, $\lambda_{em} = 589$–720 nm). **B**, **C** Left and Right CPL channel ($\lambda_{ex} = 355$ nm, $\lambda_{em} = 589$–599 nm), respectively. **D** Left-handed EDCC image (left CPL—right CPL) highlighting enantioselective predominantly lysosomal localisation. **E** Transmission image. Bottom row Λ-Eu:L1. **F**, **G**, **H**, **J** as per (**A**, **B**, **C**, **E**). **I** Left-handed EDCC image (left CPL—right CPL) highlighting enantioselective predominantly mitochondrial localisation. Scale bars = 20 μm, numbers in red are avg. Eight-bit pixel intensity values for each image region.

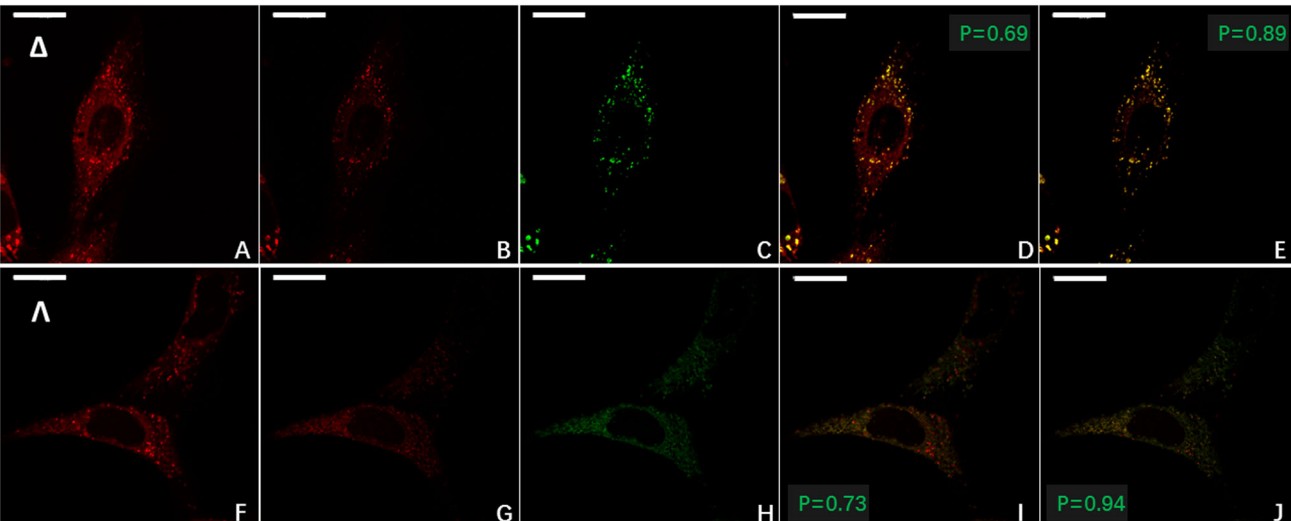

**Fig. 6 Enantioselective co-localisation of Eu:L1 with achiral commercial co-stains in live NIH 3T3 cells.** Live cell enantioselective differential chiral contrast (EDCC) co-localisation CPL-LSCM of Eu:L1 with commercial co-stains ($\lambda_{ex} = 488$ nm, 2 mW, $\lambda_{em} = 500$ to 530 nm) in NIH 3T3 (×63 1.4 NA oil objective, 96 × 96 μm FOV, 100 avg., 790 nm axial section). Top row (**A**–**E**) Δ-Eu:L1. **A** Total europium emission ($\lambda_{ex} = 355$ nm, 20 mW, $\lambda_{em} = 589$ to 720 nm). **B** Right-handed EDCC image (right CPL—left CPL) highlighting predominantly Lysosomal localisation. **C** LysoTracker[TM] Green. **D** RGB merge of (**A**) and (**C**). **E** RGB merge of **B** and **C** showing enhancement of Pearson's coefficient. Bottom row (**F**–**J**) Λ-Eu:L1. **F** as per (**A**). **G** Left-handed EDCC image (left CPL—right CPL) highlighting predominantly mitochondrial localisation. **H** MitoTracker[TM] Green. **I** RGB merge of **F** and **H**. **J** RGB merge of **G** and **H** showing enhancement of Pearson's coefficient. Scale bars = 20 μm, green numbers indicate Pearson's coefficient for each RGB merge.

To investigate further the enantioselective localisation of Λ- and Δ-Eu:L1, we repeated this experiment with the addition of LysoTracker[TM] DND-26 Green and MitoTracker[TM] FM Green dyes. From the differential CPL-LSCM images (Fig. 6), it can be seen that the Δ-enantiomer strongly overlaps with the emission from LysoTracker[TM] DND-26 Green (Pearson's-coefficient = 0.89, Fig. 6E), whereas the Λ-enantiomer strongly overlaps with that of MitoTracker[TM] FM Green (Pearson's-coefficient = 0.94, Fig. 6J). Without enantioselective differential CPL imaging, these

coefficients are significantly reduced (0.69 and 0.73, Fig. 6D, I, respectively).

This study unequivocally demonstrates and confirms that the different enantiomers Eu:L1 selectively target different cellular organelles. Enantioselective imaging is a significant milestone in live-cell optical microscopy. The above detailed simple modifications to any existing LSCM enable researchers worldwide to exploit the importance and versatility of chiral bioimaging. This could trigger a paradigm shift in the fields of molecular cell

biology and chemistry, opening new avenues in bioprobe research by enantioselectively targeting and tracking chiral species.

## Discussion

To conclude, we have presented three major advances in the intertwined disciplines of CPL spectroscopy and chiral luminescence imaging. Firstly, we have demonstrated the CPL-LSCM system, which is capable of rapidly and simultaneously acquiring diffraction-limited enantioselective chiral-contrast based CPL-differential images for the sub-cellular tracking of emissive chiral sub-cellular probes. This milestone is accompanied by the proof-of-concept demonstration of 2PE-CPL spectroscopy, showing that low-energy 2PE-CPL-LSCM is ripe for future development. Secondly, we have created a well-controlled inert CPL test target for use as a standard for enantioselective chiral-contrast imaging. This test target consists of enantiopure solutions of a suitable europium complex simultaneously drop-cast in a spatially separated manner directly onto conventional glass microscope slides. This simple test target can serve to compare and benchmark future enantioselective chiral-contrast CPL differential imaging systems that may be developed by the multi-disciplinary imaging community. Thirdly, we have imaged Λ- and Δ-enantiomers of a known chiral europium complex that localises to different organelles within mouse skin fibroblast (NIH 3T3) cells, with each enantiomer showing an excellent co-localisation profile with common lysosomal and mitochondrial tracker dyes.

We anticipate that these developments will open uncharted research avenues for fundamental studies of chiral interactions in the domains of chemistry and molecular biology. CPL-LSCM will be a vital new tool in optical microscopy aiding the development of chiral bio-probes to be taken to the next level. In material sciences, CPL-LSCM could be used for characterising 3D display technologies (e.g. emissive chiral polymers) and verification of physically unclonable stochastically micro-patterned CPL-active security inks[16–18]. Ultimately, the work presented here opens a new window into the world of chiral molecular interactions which is ripe for exploration by shedding light on the previously unexploited chiral biochemical processes that fundamentally underpin life.

## Methods

**EDCC imaging**. Microscope control and image acquisition and analysis were performed with Leica's commercial microscope software LAS-X. EDCC imaging is achieved with ImageJ software (v1.49)[31], with its built-in image calculator add-on software by subtracting one CPL channel from the other, and vice versa. The convention used herein is: left-handed enantioselective contrast = left CPL—right CPL. Right-handed enantioselective contrast = right CPL—left CPL. Images for a typical $1024 \times 1024$ pixel FOV with 10 line accumulation bidirectional scanning sequence were typically acquired in 90 s (corresponding to 10 full-frame accumulations). Confocality—axial (z) resolution—was governed by the applied pinhole diameter denoted in Airy disk units to preserve resolution and maximise light detection from the FOV.

Images were obtained using the detector saturation mode of LAS-X where each image is assessed for maximum intensity value. Each image is only recorded if no $4 \times 4$ pixel cluster (Nyquist sampling covering an area determined by the systems optical resolution, $126 \times 126$ nm at 355 nm excitation using 1.4 NA objective) possesses average intensity values of 255 on an 8-bit greyscale. This allows the employed gain of each detector to be synchronised and kept constant, so no error associated with pixel intensity saturation is included accidentally contributing to pixel uncertainties, lowering S/N and exponentially increasing the limit of detection values.

Non-live cell 8-bit average pixel chiroptical contrast value calculations were facilitated by selecting and averaging five different positions non-overlapping equal size and shape arbitrary area of the sample with respect to each enantiomer and dark background. Due to the $1024 \times 1024$ pixel size of each recorded image total FOV, this arbitrary area has been kept at a constant area of $100 \times 100$ pixels region of interest (ROI). The average maximum 8-bit greyscale pixel intensity values were determined using the LSCM's built-in LAS-X software that is employing a maximum average value ROI histogram methodology that is based on standard Gaussian distribution profiling of the average intensity values. Due to the employed methodology and the averaging nature of image acquisition and ROI calculation

the limit of detection (error associated with) 8-bit greyscale contrast value is below 1% (<3 average greyscale value on a 0–255 pixel intensity scale).

During live-cell imaging, adaptations of the above detailed chiroptical contrast greyscale value determination methodology were applied. In this case, full FOV 8-bit contrast values have been calculated using the LAS-X software. To eliminate errors associated with the number of cells occupying the FOV—in other words, the variable amount of dark 'black' background in each image, this value has been corrected with a below limit of detection value correction. In each case, the software only uses pixel for the average 8-bit greyscale intensity value determines if the intensity value of the pixel is above the value of 4 (on a 0–255 pixel intensity scale). This is determined using the total Europium emission image and the selected arbitrary ROI area selection is then kept identical throughout the imaging sequence resulting in high precision chiroptical contrast calculations.

Pearson correlation coefficients were calculated following standardised protocols using the corresponding channel of the colour split raw images using the ImageJ—JACoP plug in[33].

**Statistics and reproducibility**. Where instruments incorporating a scanning monochromator have been used (absorption, emission, and excitation spectra) each sample have been recorded and averaged as triplicate measurements. Spectra, where CCD detectors have been employed, such as two-photon cross-section determination, multiphoton and CPL spectroscopy, have been measured as an average of a thousand spectra on triplicate samples.

Microscopy images presented herein are representative images of the experiments discussed. Each experiment has been repeated in triplicates and each sample has been recorded and studied recording a minimum of five separate imaging sequences.

Imaging parameters presented, such as brightness and chiroptical contrast have been calculated on each individual imaging sequence according to the protocol detailed in the Methods and the Supplementary Information (SI) section.

**Reporting summary**. Further information on research design is available in the Nature Research Reporting Summary linked to this article.

## Data availability

All data generated and analysed, including spectra and raw microscopy images, during this study are available from the corresponding author upon request.

## Code availability

Custom codes are written in Labview2013 and Matlab2019b developed and used during this study are available from the corresponding author upon request.

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

## Acknowledgements

R.P. acknowledges support from the Royal Society University Research Fellowship URF \R\191002 and H2020-MSCA-ITN-859752 HEL4CHIROLED. R.P. and P.S. acknowledge support from BBSRC BB/S017615/1, L.M. acknowledges support from the BBSRC Discovery Fellowship BB/T009268/1. P.S. would like to thank Dr. Piotr Pander for his assistance with spin-coating. R.P. thanks Prof. Andrew Beeby for the stimulating discussions regarding CPL spectroscopy. D.P. thanks Dr. Matthieu Starck and Dr. Andrew Frawley for the synthesis of Eu complexes used herein. We are also thankful for the ongoing support of Leica Microsystems UK (Dr. Malcolm Lang, Brian Preston and Jason Lewis) and Coherent UK (Graham Wright and Blair Welsh).

## Author contributions

P.S.:[1] Co-constructed the CPL-LSCM performed all blotting and casting experiments and drafted/edited the paper. L.M.:[1] Co-constructed the CPL-LSCM, Co-designed SS-CPL and edited the paper. D.P.: Secured funding for the proof-of-concept epi-fluorescence microscope with R.P., provided lanthanide complexes for analysis and edited the paper. R.P.: Secured project funding, designed and constructed the confocal CPL microscope, performed multi-photon and live-cell experiments, and drafted/edited the paper. [1]Co-first authors can prioritise their names when adding this paper's reference to their resumes.

## Competing interests

R.P. is an inventor on patent WO2016174395A1: Light detecting apparatus for simultaneously detecting left- and right-handed circularly polarised light. The authors declare no competing interests.

## Additional information

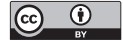

