## [Peer Review File · Nature Communications]

Circularly polarised luminescence confocal microscopy:
a live cell imaging modality to study chiral molecular
interactionsEditorial Note: Parts of this Peer Review File have been redacted as indicated to remove third-party material where no permission to publish could be obtained.

REVIEWER COMMENTS

Reviewer #1 (Remarks to the Author):

The authors describe their own original experimental apparatus for CPL imaging analysis with the use of emissive chiral probes.

However, at this stage, it is hard to clarify the scientific value of the manuscript, because there are some errors, insufficient explanation of the unit, poor resolution images in the manuscript. 
For examples, it is completely impossible to recognize some characters in Figure 1 (especially insert ones), although it involves physically important information to specify value of the experimental results.

No sub figures F-J exist in Figures 5 and 6, although it is mentioned in their captions.

No definition of GM unit anywhere.

Need to specify the definition of "GM" for broad readers in nature communications.

No explanation for relationship between top and bottom axis in Figures 1 and 3.

If bottom axis covers wavelengths from 250 to 600 nm, top axis should be from 500 to 1200 nm, because it is two photon process.

Wavelength range for top axis should not be changed freely for fair comparison, it there is no scientifically adequate reason.

The authors showed only differential images(left CPL – right CPL) in Figures 4-6.

In order to understand enhancement of enatio-selectivity for the chiral imaging analysis, summed images (left CPL + right CPL) must be informative.

After revision of these points raised above and also carefully-check of errors, I would like to read the manuscript again and provide my new scientific opinions to it.

Reviewer #2 (Remarks to the Author):

On the basis of using two previously-described europium complexes able to exhibit circularly polarized luminescence (CPL) and to act as specific probes of two different cell organelles (mitochondria and lysosomes), Pal et al. report two interesting and technologically valuable results, which are well supported by the conducted experiments. These results are:

(1) The extension of their recently developed CPL spectrometer technology to CPL Laser Scanning Confocal Microscopy (CPL-LSCM), by the demonstration of the first CPL-based differential imaging of subcellular systems (cell organelles), the latter an original idea to be explored and exploited. Interestingly, the obtained subcellular bioimages exhibit enhanced resolution (slightly higher Pearson's co-localization coefficients) when compared to those obtained by "classic" (non CPL-based) confocal microscopy using the same probes.

(2) The proof-of-concept demonstration of CPL spectroscopy by two-photon absorption (2PA), showing that low-energy 2PA-based CPL-LSCM is ripe for development.

Up to my knowledge, both results are unprecedented and constitute milestones in the fields of CPL and microscopy, deserving publication in NC.

Authors Response to Reviewers Comments

We greatly appreciate the reviewers' constructive comments and have attempted to address them in detail in the revised manuscript.

Reviewer #1 (Remarks to the Author):

Comment 1) The authors describe their own original experimental apparatus for CPL imaging analysis with the use of emissive chiral probes. However, there are some errors, insufficient explanation of the unit, poor resolution images in the manuscript.

Response to Reviewer 1 Comment 1

We have addressed all of the errors (as below, commenting on each specific point raised), and have explained the unit 'GM' used in two photon cross section determination studies.

As per the poor resolution of the images – We are often faced with this comment. All figures included in the manuscript have been submitted as high-resolution images. Each individual image has been included in its original raw state as a .TIF file with the resolution 1024 x 1024 pixels and an 8-bit colour attribution. We can only speculate that the reviewer might have commented on image resolution as read on the printed/downloaded PDF version of our submitted manuscript. PDF conversion of high-resolution images always, despite our best efforts, result in an inherent loss of resolution and compression of the figures. This is in microscopic terms often called 'Lossy image compression'. This allows the final PDF to be saved with a substantially reduced file size but as a trade off it results in poorer resolution of the figures. We are certain, and assure the reviewer, that this will be resolved in the final printed article and that the online figures that are made available will use the uncompressed high-resolution original TIF images. We will supply the original TIF images to the editor upon resubmission.

Comment 2) it is completely impossible to recognize some characters in Figure 1 (especially insert ones), although it involves physically important information to specify value of the experimental results.

Response to Reviewer 1 Comment 2

We thank the reviewer for highlighting the issues of clarity associated with the photophysical figures. On the new high resolution TIF format images submitted for figures 1, 3 and SI figure 3, to aid the reader, we have increased the font sizes and now each text caption is more easily legible.

Comment 3) No sub figures F-J exist in Figures 5 and 6, although it is mentioned in their captions.

Response to Reviewer 1 Comment 3

We thank the reviewer for spotting this mistake, despite proof-reading our manuscript multiple times this one escaped our attention. It has now been resolved in each figure legend.

Comment 4) No definition of GM unit anywhere. Need to specify the definition of “GM” for broad readers in nature communications.

Response to Reviewer 1 Comment 4

We thank the reviewer for this comment and fully agree that it should be included to aid the general reader. This has been addressed at the first time the unit [GM] is used in the text.

Its 2PE cross-section was determined to be $\sigma_2 = 142 \pm 3$ and 147 ± 3 GM ($1 \text{ GM} = 10^{-50} \text{ cm}^4 \text{ s photon}^{-1}$) for the Λ - and Δ - enantiomers in methanol (MeOH) respectively.

Comment 5) No explanation for relationship between top and bottom axis in Figures 1 and 3. If bottom axis covers wavelengths from 250 to 600 nm, top axis should be from 500 to 1200 nm, because it is two photon processes.

Wavelength range for top axis should not be changed freely for fair comparison, if there is no scientifically adequate reason.

Response to Reviewer 1 Comment 5

The top and bottom axis both refers to wavelength regions studied for both 1PE excitation and 2PE excitation. In these wavelength comparison graphs, it is common practice to display the wavelength ranges so that the shape of both 1PE excitation and 2PE excitation correspond to each other allowing the reader to determine the maximum of both 1PE and 2PE events. Such a format is used in reference 23, Figure 2, for example as included below). We must emphasize that in this ACS Nano publication, we originally presented the data as the reviewer suggest here. However, during the review process the reviewer of the above cited publication insisted that the data is presented in the style used herein. The reasoning given was the following:

‘Due to the nonlinear effect of 2PE (non-degenerate two-photon absorption) the 2PE wavelength is extremely rarely double of the 1PE absorption maximum. Due to the quadratic relationship between the intensity of the 2PE excitation and the triggering of a fluorescent event the shape of 2PE excitation is inherently always far narrower, sharper too than that of the 1PE excitation. If the data would be presented with doubling 1PE wavelength range to 2PE wavelength range both the excitation maximum and the shape would be offset to one another. This could easily lead to confusion in the reader falsely suggesting that the 2PE event is not taking place as described.’

{Redacted}

Below we show side by side comparison between the applied and suggested style of figure 1.

If both reviewer 1 and the editor feel that this 'double wavelength matched' presentation style, suggested by reviewer 1, will be more adequate to use we are happy to exchange figure 1, 3 and SI figure 3 for this new presentation style as they are already prepared. We do, however, feel that the originally chosen style makes the figure clearer and aids the general reader's appreciation for the 1PE and 2 PE comparison.

Comment 6) The authors showed only differential images(left CPL – right CPL) in Figures 4-6. In order to understand enhancement of enantio-selectivity for the chiral imaging analysis, summed images (left CPL + right CPL) must be informative.

Response to Reviewer 1 Comment 6

For each figure the total emission ($I_{\text{left}} + I_{\text{right}}$) and CPL or differential chiroptical contrast image ($I_{\text{left}} - I_{\text{right}}$) is shown, alongside the individually selected I_{left} and I_{right} channel images. The requested 'summed' (left CPL + right CPL) therefore is already included in these figures as the first row of each imaging sequence (total emission). The emission in this case has been collected for the wavelength range (total synthesised Eu(III) emission) 589-720 nm including all the ΔJ_{1-4} Eu(III) transitions. (Please note that ΔJ_0 is CPL inactive, due to its pure electric dipole allowed character.)

The overall CPL signal in spectral form for each enantiomer – as per equations above – are mirror images of one another (as depicted in figure 1 and 3 section C). In our instrumental setup individual imaging channels are dedicated to capture both pure left and right-handed emitted circularly polarised light over a suitable ΔJ transition that is all one - positive (Δ enantiomer) or negative (Λ enantiomer) - CPL sign with broad wavelength span and high g_{lum} value. This after pixel-by-pixel total image subtraction give rise to the absolute differential chiroptical images for each enantiomer.

Importantly, the overall cellular localisation profile of each enantiomer does not change whether all Eu(III) transition is collected or only one photophysically favoured ΔJ_x transition is selected.

To demonstrate the validity of the above with respect to Figure 5 the left image below is the included total Eu(III) emission profile for the delta enantiomer (figure 5A wavelength range 589-720 nm) whilst the right image below is the requested sum of 5B (Left CPL channel, wavelength range 589-599 nm) and 5C (Right CPL channel, wavelength range 589-599 nm). The overlap coefficient – Pearson's coefficient – of the two images is 1, meaning that they are identical, as reasoned above.

This perfect overlap is visualised by the unified yellow colour on the image on the right below where the images 5A (left, kept in red colour) and the sum of figure 5B+C (middle, pseudo-coloured green) have been overlaid, as an RGB merge.

This scientifically underpins the previously stated, that the overall cellular localisation profile of each enantiomer is wavelength independent in the emissive range and does not change whether all Eu(III) transition is collected or only one photophysically favoured ΔJ_x transition is selected.

However, it is important to consider two key points with respect to CPL microscopy:

First and foremost, for achieving the highest contrast (intensity value difference) vs. image acquisition time ratio in chiroptical imaging that is of extreme importance during live-cell imaging, we have selected the Eu(III) transition (ΔJ_1) that has the highest g_{lum} to be studied by the applied band pass filter (BP594/10), as for figure 3C this value is -0.26 for the delta enantiomer. This is due to the predominantly magnetic dipole allowed nature of the ΔJ_1 transition.

But most importantly, if we were to study the difference in chiroptical contrast using a filter covering the total Eu(III) range 589-720 (as per figure 3C), the total observable chiroptical contrast would be – looking at the Delta (red line) enantiomer again –

$$\text{Total } g_{lum} = g_{lum}(\Delta J_1) + g_{lum}(\Delta J_2) + g_{lum}(\Delta J_3) + g_{lum}(\Delta J_4) \quad (-0.26) + 0.11 + 0.04 + 0.01 = \mathbf{-0.1}$$

(as opposed to - 0.26)

Please note that $\Delta J_{3 \text{ and } 4}$ are averaged g_{lum} values, as the transitions are non-single sign transitions.

If the total Eu(III) emission intensity is used for this compound's enantiomers, the measurement would on average take 2.5 times longer to collect the same contrast difference for CPL in each channel, with an inherently $2.5^2 = 6.25$ times lower S/N ratio (square relationship between $t_{accumulation}$ and S/N). This would prolong image collection and result in image blurring (apparent resolution loss) during live cell imaging, due to time dependent natural homeostatic rearrangement and movement of the intracellular architecture.

Comment 7) After revision of these points raised above and also carefully-check of errors, I would like to read the manuscript again and provide my new scientific opinions to it.

Response to Reviewer 1 Comment 7

We hope that our response to scientific comments 1 – 6 will be found to be satisfactory by the reviewer.

Reviewer #2 (Remarks to the Author):

Comment 1) On the basis of using two previously-described europium complexes able to exhibit circularly polarized luminescence (CPL) and to act as specific probes of two different cell organelles (mitochondria and lysosomes), Pal et al. report two interesting and technologically valuable results, which are well supported by the conducted experiments. These results are:

(1) The extension of their recently developed CPL spectrometer technology to CPL Laser Scanning Confocal Microscopy (CPL-LSCM), by the demonstration of the first CPL-based differential imaging of subcellular systems (cell organelles), the latter an original idea to be explored and exploited. Interestingly, the obtained subcellular bioimages exhibit enhanced resolution (slightly higher Pearson's co-localization coefficients) when compared to those obtained by "classic" (non CPL-based) confocal microscopy using the same probes.

(2) The proof-of-concept demonstration of CPL spectroscopy by two-photon absorption (2PA), showing that low-energy 2PA-based CPL-LSCM is ripe for development. Up to my knowledge, both results are unprecedented and constitute milestones in the fields of CPL and microscopy, deserving publication in NC.

Response to Reviewer 1 Comment 7

We are delighted that the reviewer shares our excitement for the microscope design and acknowledges the potential and step-changing nature of it for optical microscopy and associated research disciplines.

We hope that these corrections and modifications to the original text satisfy the reviewers.

Sincerely,

Robert Pal

REVIEWER COMMENTS

Reviewer #1 (Remarks to the Author):

The authors amended points which were raised to the first manuscript.

With regard to comment #2, it is still hard to distinguish characters for horizontal and vertical axes in the inset of the Figure 1A and 3A,
 I_{2PE} (a.u.), P_{in} (mW).

Probably, it is also better to more directly state in the Figure caption.

With regard to comment #5, The author stated historical reasons of the choice of wavelength regions for top and bottom axes.

I know and understand realistic situation that "the 2PE wavelength is extremely rarely double of the 1PE absorption maximum".

Offset and shape modification between 2PE and 1PE process, however, involves important physical meaning behind.

Rather than freely selecting wavelength region, simply doubled axis should be appropriate to understand the property of the chiral emitters.

But not to confuse the broad but scientific readers in nature communications, sufficient explanation of the offset and shape modification must be required.

At line 168 on page 6, there is a statement of "selected by rotating a linear polariser". Purpose (meaning) and also location of the linear polarizer is not clear to me, although "two" polarizers are equipped in the system in Fig. 2.

It is probably better to mention the previous work of (single photon induced) CPL luminescence imaging (inset of Fig 3 of your ref. 20).

In Figure 4, image contrast was discussed quantitatively. Which part (area? Cross-section?) was used for the evaluation. It should be clearly state a method of the evaluation. Intensity scale (color scale) for the images or line profile of the images may be helpful to see the quality of the microscopic measurement.

In addition, quantitative evaluation of the detection limit/sensitivity must be clearly discussed somewhere to show value of this methodology and understand its limitations.

If the points above were considered carefully, value of the manuscript must be improved.

Authors Response to Reviewers Comments

We greatly appreciate the reviewers second round of comments and have attempted to address them in detail in the revised manuscript.

Reviewer #1 (Remarks to the Author):

The authors amended points which were raised to the first manuscript.

Thank you we are delighted that our response and amendment of the manuscript accordingly were satisfactory.

Comment 1) With regard to comment #2, it is still hard to distinguish characters for horizontal and vertical axes in the inset of the Figure 1A and 3A,

I_{2PE} (a.u.), P_{2PE} (mW).

Probably, it is also better to more directly state in the Figure caption.

With regard to comment #5, The author stated historical reasons of the choice of wavelength regions for top and bottom axes.

I know and understand realistic situation that “the 2PE wavelength is extremely rarely double of the 1PE absorption maximum”.

Offset and shape modification between 2PE and 1PE process, however, involves important physical meaning behind.

Rather than freely selecting wavelength region, simply doubled axis should be appropriate to understand the property of the chiral emitters.

But not to confuse the broad but scientific readers in nature communications, sufficient explanation of the offset and shape modification must be required.

Response to Reviewer 1 Comment 1

Taking the reviewers comments on board we have redesigned, relabelled and redistributed figure 1 and 3 and SI3, which now hopefully will be found satisfactory and visually pleasing. A clarification on the 2PE excitation maximum with respect to wavelength and shape has also been included in the text to aid the board readership of Nature Communication.

To gain full appreciation, we must emphasize that due to the nonlinear effect of 2PE (non-degenerate two-photon absorption) the 2PE wavelength is often lower and extremely rarely double of the 1PE absorption maximum. Due to the quadratic relationship between the intensity of the 2PE excitation and the triggering of a fluorescent event the shape of 2PE excitation is inherently always far narrower, sharper too than that of the 1PE excitation.

Comment 2) At line 168 on page 6, there is a statement of “selected by rotating a linear polariser”.

Purpose (meaning) and also location of the linear polarizer is not clear to me, although “two” polarizers are equipped in the system in Fig. 2.

Response to Reviewer 1 Comment 2

We have rephrased the sentence in question and cited reference 15 where the reader can gain full appreciation of our instrumental design that is identical in optical arrangement to our 2020 Nature Communication work on our novel fast dual CCD time-resolved CPL-spectrometer design.

The two linear polarised light states generated (horizontal and vertical polarisation) corresponding to left or right CPL are selected by a carefully aligned linear polariser, housed in a high-precision computer-controlled rotation mount.¹⁵ Emission intensity of each pixel is quantified in a conventional LSCM scanned manner by a dedicated high sensitivity avalanche photodiode pair.

Comment 3) It is probably better to mention the previous work of (single photon induced) CPL luminescence imaging (inset of Fig 3 of your ref. 20).

Response to Reviewer 1 Comment 3

We have deliberately constructed our manuscript to provide the reader with a full appreciation towards how our work have been evolved from our previous independent work and the directions we have taken towards live cell CPL imaging. Our pioneering single photon CPL epi-fluorescence imaging work (Ref. 9, 2016) is extensively featured in this manuscript and discussed at the beginning of the Results section for the reader to gain deep understanding on its pivotal role in the instrumental evolution aided by our work Ref. 15 (2020) of our current work presented herein. Both Ref 9 and 15 is pivotal to this work and featured extensively, urging the reader to familiarise themselves with them to gain full appreciation of the work presented herein.

We have cited reference 20 (2016) to highlight the strongest CPL signal generated by chiral Eu containing supramolecular polymer compounds to date. We feel that in the context of the work reported herein we have included and praised this important milestone work (ref. 20) where felt it is most appropriate with its superior engineered gLum values reported.

The work indeed as quoted by the reviewer includes a sub figure in Figure 3. Importantly, unlike our work presented here it does not feature the all-important simultaneous parallel detection of two different enantiomers of a strongly emitting CPL active compound.

{Redacted}

It is also not clear from the paper, but we assume this image was taken of the studied aggregates in solution in a glass vial. This could explain the equally brightness 'orange' bands on the bottom of A and B where reflection of the emitted CPL light from the bottom of the vial is randomized due to reflection. The top parts of each section (A and B) in 'CPL brightness' is eye differentiable, due to the above mentioned superior engineered gLum values of the polymer and the subfigure is based on

intensity selected by a bandpass filter hence, we have already dedicated a small paragraph of discussing its importance in our 2020 Nat. Chem. Rev. paper on CPL security inks.

However, on an instrumental point of view that is the key importance of our work presented herein the work presented in Ref 20 does not provide the reader with a detailed experimental set up on how this image have been achieved experimentally. The only experimental detail was included in the figure legend: ***Inset: Visible luminescence image of Cs+[Eu((+)-hfbc)4]- with a band path filter (592 nm) and (A) right-, (B) left-circularly polarized filter.***

In order to gain full appreciation of its significance and the brightness difference displayed in the figure insert the authors should have also specified the exact specification (FWHM and band width) of the 592 band pass filter that was used with the 'circularly polarised filter' (we assume from the paper that it is quarter waveplate and carefully aligned linear polariser). We have used an identical set up—developed independently by us - that we have discussed extensively in our 2016 work with scheme 1 and 3 in both DSLR based and microscopy based format paving the way towards handheld authentication instrumentation and the CPL-LSCM live cell microscope presented herein allowing simultaneous differential chiroptical imaging of compounds with modest gLum values in live cells.

{Redacted}

Comment 4) In Figure 4, image contrast was discussed quantitatively. Which part (area? Cross-section?) was used for the evaluation. It should be clearly state a method of the evaluation. Intensity scale (color scale) for the images or line profile of the images may be helpful to see the quality of the microscopic measurement.

In addition, quantitative evaluation of the detection limit/sensitivity must be clearly discussed somewhere to show value of this methodology and understand its limitations.

If the points above were considered carefully, value of the manuscript must be improved.

Response to Reviewer 1 Comment 4

We thank the reviewer for the suggestion. We have now included a dedicated post image processing calculation section into to SI detailing how each pseudo coloured imaged (colour coding is necessary for greater appreciation of differential chiroptical imaging) 8 bit grayscale value (0–255 scale) has been calculated and processed including the limit and error of detection of our system.

Full technical details of the CPL-LSCM system and the applied image processing methodology are provided in the supporting information.

IN SI:

Images were obtained using the detector saturation mode of LAS-AF where each image is assessed for maximum intensity value. Each image is only recorded if no 4 x 4 pixel cluster (Nyquist sampling covering an area determined by the systems optical resolution, 126 x 126 nm at 355 nm excitation using 1.4 NA objective) possesses average intensity values of 255 on an 8 bit greyscale. This allows the employed gain of each detector to be synchronised and kept constant, so no error associated with pixel intensity saturation is included accidentally contributing to pixel uncertainties, lowered S/N and exponentially increasing limit of detection values.

Non live cell 8-bit average pixel chiroptical contrast value calculations were facilitated by selecting and averaging 5 different position non-overlapping equal size and shape arbitrary area of the sample with respect to each enantiomer and dark background. Due to the 1024 x 1024 pixel size of each recorded image total field of view (FOV) this arbitrary area has been kept at a constant area of 100x 100 pixels region of interest (ROI). The average maximum 8-bit grayscale pixel intensity values were determined using the LSCM's built in LAS-AF software that is employing a maximum average value ROI histogram methodology that is based on standard Gaussian distribution profiling of the average intensity values. Due to the employed methodology and the averaging nature of image acquisition and ROI calculation the limit of detection (error associated with) 8-bit grayscale contrast value is below 1% (<3 average grayscale value on a 0 – 255 pixel intensity scale).

During live-cell imaging, adaptations of the above detailed chiroptical contrast greyscale value determination methodology was applied. In this case full field of view (FOV) 8-bit contrast value have been calculated using the LAS-AF software. To eliminate errors associated with the number of cells occupying the FOV – in other words the variable amount of dark 'black' background in each image, this value has been corrected with a below limit of detection value correction. In each case the software only uses pixel for the average 8-bit grayscale intensity value determination if the intensity value of the pixel is above the value of 4 (on a 0 – 255 pixel intensity scale). This is determined using the total Europium emission image and the selected arbitrary ROI area selection is then kept identical throughout the imaging sequence resulting in high precision chiroptical contrast calculations.

We hope that these second round of corrections and modifications to the original revised text satisfy the reviewers.

Sincerely,

Robert Pal

REVIEWERS' COMMENTS

Reviewer #1 (Remarks to the Author):

I have carefully read the author's rebuttal, amended manuscript and SI. I believe that almost all the points that I raised in the previous communication have been properly revised by the authors and a current form of the revised manuscript attracts broad readership in Nature Communications. The value of this work will be recognized by many readers in the field of life science and many other relating fields.